# Ultralarge anti-Stokes lasing through tandem upconversion

Tianying Sun[1,2,3,7], Bing Chen [1,3,7], Yang Guo[1,3], Qi Zhu[1,3], Jianxiong Zhao[1,3], Yuhua Li[4], Xian Chen[5], Yunkai Wu[6], Yaobin Gao[2], Limin Jin [6✉], Sai Tak Chu [4✉] & Feng Wang [1,3✉]

Coherent ultraviolet light is important for applications in environmental and life sciences. However, direct ultraviolet lasing is constrained by the fabrication challenge and operation cost. Herein, we present a strategy for the indirect generation of deep-ultraviolet lasing through a tandem upconversion process. A core–shell–shell nanoparticle is developed to achieve deep-ultraviolet emission at 290 nm by excitation in the telecommunication wavelength range at 1550 nm. The ultralarge anti-Stokes shift of 1260 nm (~3.5 eV) stems from a tandem combination of distinct upconversion processes that are integrated into separate layers of the core–shell–shell structure. By incorporating the core–shell–shell nanoparticles as gain media into a toroid microcavity, single-mode lasing at 289.2 nm is realized by pumping at 1550 nm. As various optical components are readily available in the mature telecommunication industry, our findings provide a viable solution for constructing miniaturized short-wavelength lasers that are suitable for device applications.

[1] Department of Materials Science and Engineering, City University of Hong Kong, 83 Tat Chee Avenue, Hong Kong SAR, China. [2] School of Chemical Engineering and Technology, Sun Yat-sen University, Zhuhai 519082, China. [3] City University of Hong Kong Shenzhen Research Institute, Shenzhen 518057, China. [4] Department of Physics, City University of Hong Kong, 83 Tat Chee Avenue, Hong Kong SAR, China. [5] College of Materials Science and Engineering, Shenzhen University, Shenzhen 518060, China. [6] State Key Laboratory on Tunable laser Technology, Ministry of Industry and Information Technology Key Lab of Micro-Nano Optoelectronic Information System, Harbin Institute of Technology (Shenzhen), Shenzhen 518055, China. [7] These authors contributed equally: Tianying Sun, Bing Chen. ✉email: jinlimin@hit.edu.cn; saitchu@cityu.edu.hk; fwang24@cityu.edu.hk

Luminescent materials that convert excitation photons into prescribed emissions are at the core of many photonics technologies such as varicolored displays and programmable photoactivation[1,2]. Amongst various luminescence processes, photon upconversion characterized by high-energy emission upon excitation of lower-energy photons is of exceptional interest. Upconversion primarily takes advantage of lanthanide-doped materials, in which the stepwise excitation through the energy levels of the lanthanide activators results in visible and ultraviolet emissions by successive absorption of multiple near-infrared photons[3–7]. The unique upconversion process has enabled a diversity of applications ranging from bioimaging to solar energy conversion and optical storage[8–13]. In particular, upconversion is considered as a promising solution to generating short-wavelength lasing by pumping with longer-wavelength light sources that are more readily acquired[14,15].

Frequency upconversion holds potential for cost-effective construction of miniaturized deep-ultraviolet (UV) emission devices that find enormous medical and industrial applications, such as microbial sterilization and biomedical instrumentation[16–19]. However, the implementation of such a technique has been constrained by the limited spectral tunability of upconversion, which occurs in special lanthanide ions comprising fixed sets of energy levels. For example, one important class of light sources are lasers operating in the telecommunication wavelengths (1260 to 1675 nm)[20,21], which are extensively used in fiber-optic communication and photonic circuits because of minimal optical attenuation, ready accessibility in various forms, and low cost for device fabrication. In addition, the wavelengths fall in the second near-infrared window (NIR-II) that is favorable for high-resolution in vivo bioimaging owing to maximal tissue transparency and minimal autofluorescence[22,23]. However, only a small number of $Er^{3+}$-sensitized materials are capable of upconverting excitation light in this wavelength range, which displays dominated $Er^{3+}$ emissions across a limited spectrum[24–28]. It remains a daunting challenge to achieve deep-UV emission by excitation in the telecommunication wavelengths.

To expand the spectral tunability of upconversion, herein we propose a domino upconversion (DU) scheme, in which energy amassed in one upconversion course triggers another succeeding upconversion process (Fig. 1). By a tandem combination of $Er^{3+}$- and $Tm^{3+}$-based upconversion in a core–shell–shell nanostructure, deep-ultraviolet emission is realized by excitation at 1550 nm with an ultralarge anti-Stokes shift of up to 1260 nm. We systematically investigate the energy cascade processes in the core–shell–shell nanostructures and demonstrate deep-ultraviolet lasing at 289.2 nm through the DU scheme by excitation at the telecommunication wavelength.

## Results

**Synthesis and characterization.** As a proof-of-concept experiment, we constructed a $NaYF_4$:Yb/Tm@$NaErF_4$:Ce @$NaYF_4$ core−shell−shell nanoparticle with the $Tm^{3+}$- and $Er^{3+}$-based upconversion processes separately incorporated into the core and interlayer of the nanoparticle, respectively (Fig. 2a). The outermost shell of $NaYF_4$ was designed to protect the nanoparticle against surface quenching[29,30]. The spatial separation of the dopant ions was intended to minimize the cross-talk between different upconversion processes[31,32], which were independently optimized in their respective doping domains. To facilitate the DU process through interfacial energy transfer, we also employed high concentrations of $Er^{3+}$ and $Yb^{3+}$ dopants that are highly resistant to concentration quenching (Fig. 2a)[25,33,34].

The nanoparticles were synthesized by a layer-by-layer epitaxial growth protocol[35], which involved the preparation of $NaYF_4$:Yb/Tm core nanoparticle followed by the epitaxial growth of the $NaErF_4$:Ce interlayer and the $NaYF_4$ shell (Supplementary Fig. 1a). Figure 2b shows the high-angle annular dark-field scanning transmission electron microscopy (HAADF-STEM) image of the sample, revealing the highly uniform size and morphology of nanoparticles, with a distinguished Z-contrast between the $NaErF_4$:Ce (30%) interlayer and outermost $NaYF_4$ layer. The high-resolution transmission electron microscopy (HR-TEM) and powder X-ray diffraction (XRD) measurements further confirmed the high crystallinity of the nanoparticles with a single hexagonal phase (Fig. 2c, d and Supplementary Figs. 1b, 2).

**Ultralarge anti-Stokes emission through DU.** We next measured the emission spectrum of the nanoparticles under 1550 nm excitation. The nanoparticles were deposited on the top of a waveguide structure that functioned as the excitation source (Fig. 2e). The waveguide circuit was semi-buried in a $SiO_2$ substrate, with the top surface exposed to contact the nanoparticles (Supplementary Fig. 3a, b). Owing to its small dimension, the waveguide structure spatially confines the incident light and thus enhances the power density of the excitation field[36]. In a specific case, we estimated a

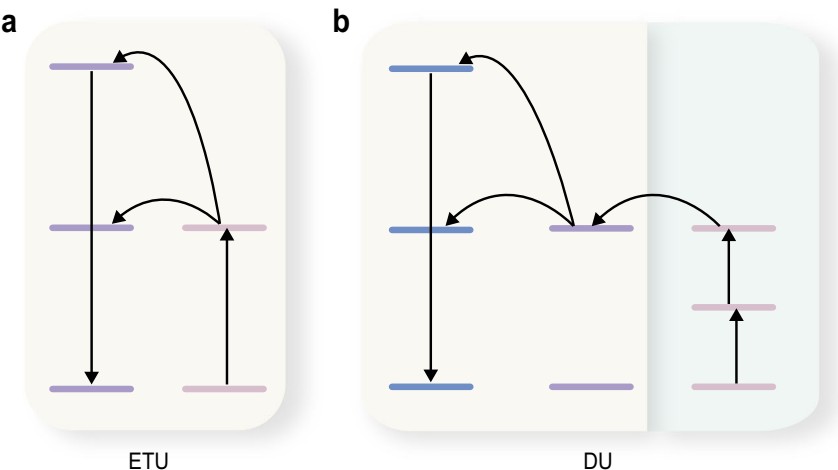

**Fig. 1 Comparison of the conventional energy transfer upconversion (ETU) and the proposed domino upconversion (DU) processes. a** In an ETU process, the excitation energy is only amassed in one type of lanthanide upconverting ion. **b** In a DU process, the excitation energy amassed in one upconverting ion triggers energy amassment in a second upconverting ion, leading to an ultralarge anti-Stokes shift.

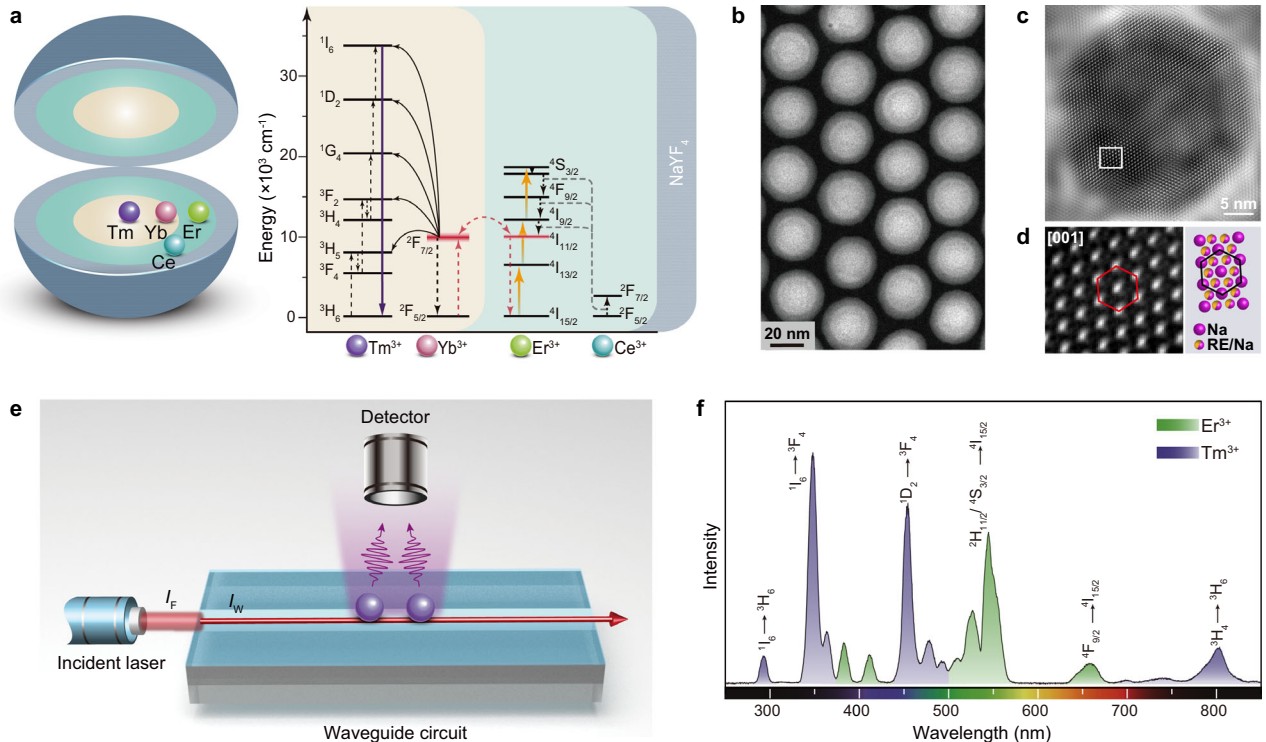

**Fig. 2 Ultralarge anti-Stokes emission through DU in core–shell–shell nanoparticles. a** Schematic design of a $NaYF_4$:Yb/Tm@$NaErF_4$:Ce@$NaYF_4$ core–shell–shell nanoparticle for DU (left panel) and proposed energy transfer mechanism in the nanoparticle. **b** HAADF-STEM image of the $NaYF_4$:Yb/Tm@$NaErF_4$:Ce@$NaYF_4$ nanoparticles highlighting the layered structure. **c** Digitally processed high-resolution TEM image of a $NaYF_4$:Yb/Tm@$NaErF_4$:Ce@$NaYF_4$ nanoparticle showing the single crystalline nature. **d** An enlarged view of the selected area in **c**, indicated by a white box, showing the hexagonal structure of the lattice in accord with the $NaYF_4$ crystal (right panel). **e** Schematic illustration of the waveguide circuit for excitation of upconversion nanoparticles. Due to the convergence of the laser beam, the power density in the waveguide circuit ($I_W$) was amplified relative to that in the incident fiber ($I_F$). **f** Emission spectrum of the $NaYF_4$:Yb/Tm@$NaErF_4$:Ce@$NaYF_4$ nanoparticles by excitation of the waveguide circuit at 1550 nm with a high-power density of 2073 kW cm$^{-2}$.

high excitation power density of 2073 kW cm$^{-2}$ at an input power of 311 mW (Supplementary Fig. 3c). The emission spectrum consists of characteristic emission peaks of $Tm^{3+}$ that can be assigned to the $^1I_6 \rightarrow {}^3H_6$ and $^3F_4$ (290 and 347 nm), $^1D_2 \rightarrow {}^3H_6$ and $^3F_4$ (362 and 453 nm), $^1G_4 \rightarrow {}^3H_6$ and $^3F_4$ (478 and 649 nm), and $^3H_4 \rightarrow {}^3H_6$ (803 nm) transitions, respectively (Fig. 2f). The observation of strong upconversion emission in the short-ultraviolet wavelength region suggests an efficient $Tm^{3+}$ upconversion sensitized by $Er^{3+}$. Note that the Yb/Tm-doped upconversion layer alone does not respond to the 1550 nm excitation (Supplementary Fig. 4).

It is worth noting that the inclusion of $Ce^{3+}$ dopants in the $NaErF_4$ layer is essential for achieving the DU process. Figure 3a compares the emission spectra of nanoparticles without and with $Ce^{3+}$ dopants in the interlayer, which revealed substantial attenuation of $Tm^{3+}$ emissions in the absence of $Ce^{3+}$. The $Ce^{3+}$ ions contributed to the DU by inhibiting high-order upconversion in $Er^{3+}$ ions through cross-relaxation (Supplementary Fig. 5), which resulted in a preferential population of the $^4I_{11/2}$ state[28]. A large $^4I_{11/2}$ population facilitated energy transfer to $Yb^{3+}$ ions and subsequent upconversion in the $Tm^{3+}$ ions (Fig. 3b, c and Supplementary Fig. 6a). Without the $Ce^{3+}$ dopants, the $Er^{3+}$ ions were straightforwardly excited to the higher-lying excited states, followed by radiative transitions to the ground state that gave rise to the dominated emission of $Er^{3+}$ ions (Supplementary Fig. 6b).

The DU process is strongly affected by the content of $Ce^{3+}$ ions. By correlating the emission intensity with $Ce^{3+}$ concentration in the interlayer, the optimal $Ce^{3+}$ doping concentration was determined to be 30% (Supplementary Fig. 7b). The reduction of

upconversion emission at substantially high $Ce^{3+}$ concentration (>30%) is partially attributed to the large lattice mismatches between the core/shell components, which resulted in nonuniform epitaxial growth processes (Supplementary Fig. 7a)[37–39].

To substantiate the role of $Ce^{3+}$ ions in the selective quenching of $Er^{3+}$ ions, we compared the visible and NIR emissions of $NaYF_4$@$NaErF_4$:Ce@$NaYF_4$ nanoparticles with and without $Ce^{3+}$ dopants (Supplementary Fig. 8). $Yb^{3+}$ and $Tm^{3+}$ ions were removed to avoid disturbance to the $Er^{3+}$ emission. As anticipated, we observed enhancement of the NIR emission in the $Er^{3+}$ ions at the expense of the visible emissions due to the inclusion of $Ce^{3+}$ dopants (Fig. 3d). Furthermore, the decay times of the $^4S_{3/2}$ and $^4F_{9/2}$ states of the $Er^{3+}$ ions were shortened by the $Ce^{3+}$ dopants, indicating a nonradiative energy transfer from $Er^{3+}$ to $Ce^{3+}$ ions (Supplementary Fig. 9).

It is worth noting that a high excitation power density is also essential for achieving the DU process[40,41]. We observed that the $Er^{3+}$ emission at around 980 nm ($^4I_{11/2} \rightarrow {}^4I_{15/2}$) was quenched by $Ce^{3+}$ dopants at low excitation powers (Fig. 3d, bottom panel). Correspondingly, DU emission in $Tm^{3+}$ ions was also quenched by $Ce^{3+}$ under low-power excitation (Fig. 3e and Supplementary Fig. 10). The results were ascribed to $^4I_{11/2} \rightarrow {}^4I_{13/2}$ cross-relaxation in $Er^{3+}$ ions induced by $Ce^{3+}$. A high excitation power promotes the $^4I_{13/2} \rightarrow {}^4I_{9/2}$ excitation process, which subsequently enhances population in the $^4I_{11/2}$ state through the $^4I_{9/2} \rightarrow {}^4I_{11/2}$ cross-relaxation (Fig. 3f). The numerical simulations based on rate equations confirmed that $Ce^{3+}$ dopants increase the population in the $^4I_{11/2}$ state of $Er^{3+}$ only at high excitation powers (Fig. 3g).

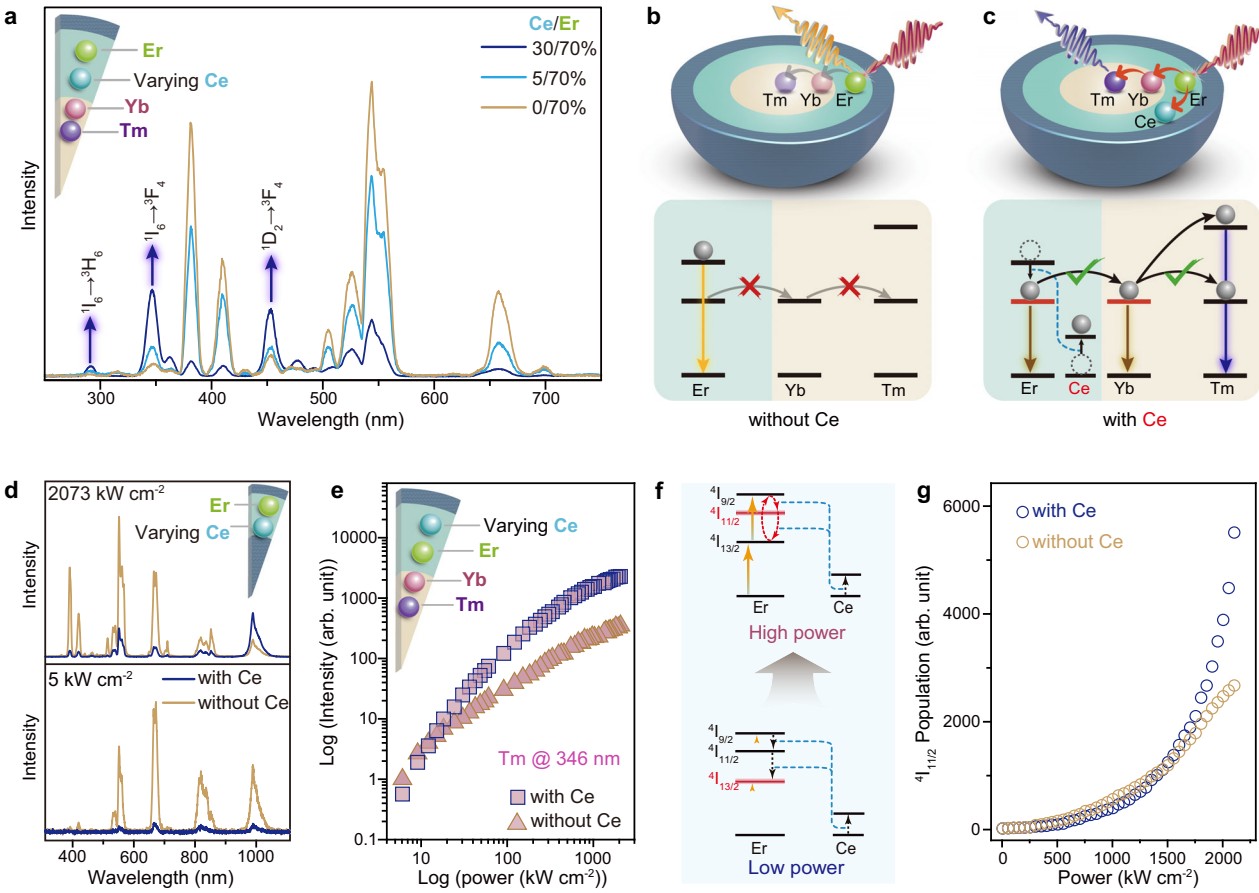

**Fig. 3 Mechanistic investigation of Ce³⁺-induced cross-relaxation. a** Emission spectra of the NaYF₄:Yb/Tm@NaYF₄:Ce/Er@NaYF₄ nanoparticles under excitation of 1550 nm at 2073 kW cm⁻² as a function of Ce³⁺ doping concentration in the interlayer. **b, c** Proposed energy transfer pathways in the NaYF₄:Yb/Tm@NaYF₄:(Ce/)Er@NaYF₄ nanoparticles without and with Ce³⁺ dopants, respectively. **d** Emission spectra of the NaYF₄@NaYF₄:(Ce/)Er@NaYF₄ nanoparticles without and with Ce³⁺ dopants under 1550 nm excitation at high (2073 kW cm⁻²) and low (5 kW cm⁻²) powers, respectively. **e** Emission intensity at 346 nm (Tm³⁺) as a function of excitation power density in the NaYF₄:Yb/Tm@NaYF₄:(Ce/)Er@NaYF₄ nanoparticles without and with Ce³⁺ dopants, respectively. **f** Schematic illustrations of excitation power-dependent preferential population of energy levels in Er³⁺ ions through cross-relaxation with Ce³⁺ dopants. **g** Simulated populations of ⁴I₁₁/₂ energy levels as a function of excitation power density in NaYF₄:(Ce/)Er without and with Ce³⁺ dopants, respectively. Mechanistic calculations by formulating the rate equations as in the Supplementary Methods.

In a further set of experiments, we demonstrate the critical role of Yb³⁺ in mediating the energy transfer from the Er³⁺ to the Tm³⁺ ions across the core/shell interface. When the Yb³⁺ ions in the core level were replaced by optically inert Lu³⁺ ions (Supplementary Fig. 11), the Tm³⁺ emission was hardly detected even at a high excitation power density of 2073 kW cm⁻² (Fig. 4a). The observation was ascribed to the large physical separation between the Er³⁺ and Tm³⁺ ions. Owing to the low dopant concentration of Tm³⁺ (1%), their average distance from the core/shell interface was too far for the energy transfer to proceed. The introduction of a high concentration (40%) of Yb³⁺ ions created an energy conduit to the Tm³⁺ ions by forming a network of the Yb³⁺ lattice, which permits fast energy migration over a long distance[42,43].

Yb³⁺ ions facilitated the energy extraction from the interlayer also due to their relatively large absorption cross-sections (~10⁻²⁰ cm²) and resonant energy level with Er³⁺ donors[44], which resulted in a long critical distance of energy transfer. Our control experiments revealed that the Yb³⁺ mediated energy transfer can still proceed when the Er/Ce shell was isolated from the Yb/Tm core by a NaYF₄ spacing layer of 2.5 nm (Fig. 4b and Supplementary Fig. 12). The energy transfer distance is appreciably larger than that observed for other ionic systems such as Gd³⁺ and Tb³⁺ (~1.1 nm)[45].

The core−shell−shell structure is also essential for achieving the DU process. As we homogeneously doped all the lanthanide ions in the core layer of a NaYF₄:Yb/Tm/Er/Ce@ NaYF₄ core−shell nanoparticle (Fig. 4c, d and Supplementary Fig. 13), the overall emission was rather weak and the Tm³⁺ emission can hardly be detected. The result was ascribed to extensive and uncontrollable energy exchange interactions among the Yb³⁺, Tm³⁺, Ce³⁺, and Er³⁺ ions, which resulted in significant dissipation of excited energy. Note that the quenching processes in the quadruply-doped system were too strong to be alleviated by high-power excitation in our experiments (Fig. 4c).

**Deep-UV lasing through DU**. The laser characteristics of the NaYF₄:Yb/Tm@NaErF₄:Ce@NaYF₄ nanoparticles were examined under free-space excitation of a 1550 nm pulse laser with 6 ns frequency duration and 10 Hz repetition rate. The as-synthesized nanoparticles were incorporated into a toroidal microresonator as the laser cavity (Fig. 5a), which supports whispering gallery mode at the internal boundaries of the nanoparticle-doped microtoroidal resonator (Fig. 5b)[46,47]. Partly owing to the high processability and small size of the upconversion nanoparticles, the composite microcavity displayed a uniform size and smooth surface (Fig. 5c). Correspondingly, a high-quality factor (Q-factor) of about 2 × 10⁵ was determined by assessing the transmission characteristic of a 1550 nm

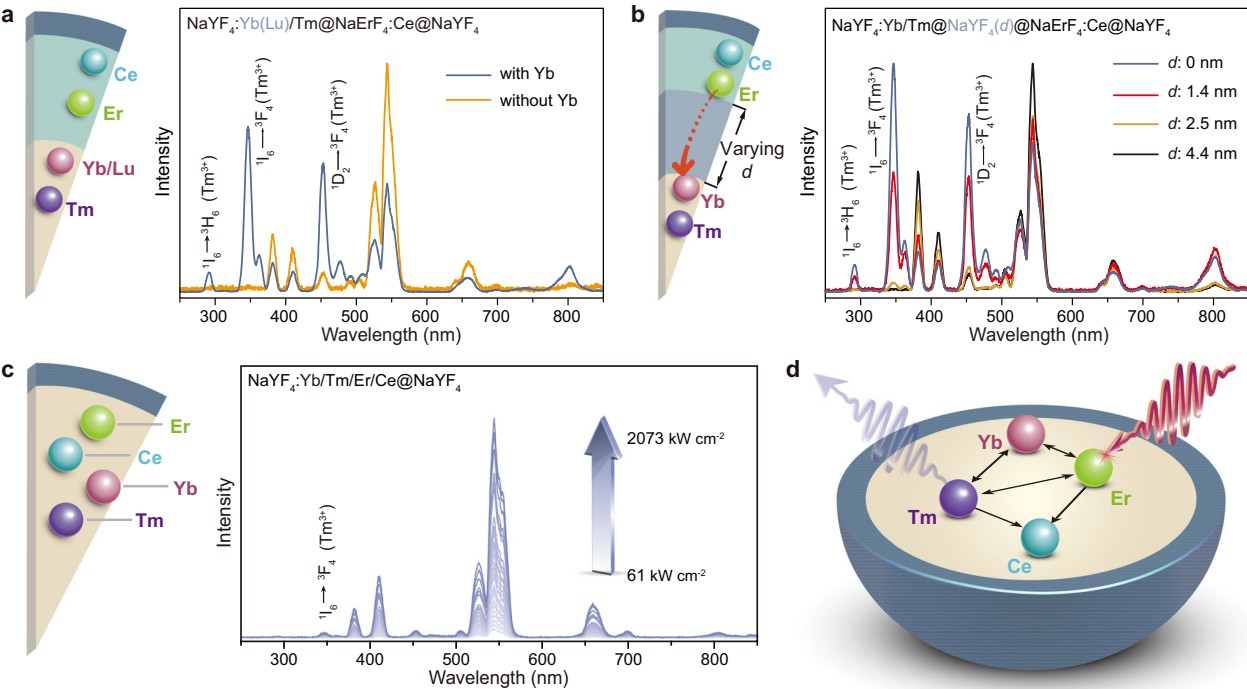

**Fig. 4 Mechanistic investigation of energy transfer in the DU process. a** Emission spectra of NaYF$_4$:Yb(Lu)/Tm@NaErF$_4$:Ce@NaYF$_4$ nanoparticles under excitation of 1550 nm at 2073 kW cm$^{-2}$, demonstrating the necessity of Yb$^{3+}$ ions for mediating energy transfer across the core/shell interface. **b** Emission spectra of NaYF$_4$:Yb/Tm@@NaYF$_4$(d)@NaErF$_4$:Ce@NaYF$_4$ nanoparticles by 1550 nm excitation, verifying involvement of interlayer energy transfer in the DU process. **c** Power-density-dependent emission spectra of NaYF$_4$:Yb/Tm/Er/Ce@NaYF$_4$ nanoparticles by 1550 nm excitation, necessitating the use of core–shell–shell design for obtaining high-efficiency DU. **d** Schematic of uncontrollable energy exchange interactions in the homogeneously doped nanoparticle.

band diode laser (Santec, TSL-710) that was coupled to the microresonator through a tapered fiber (Fig. 5d).

To examine the lasing action, the emission spectra of a typical microresonator with a 17-μm diameter were recorded as a function of pump power. As shown in Fig. 5e, a sharp emission peak (linewidth < 0.05 nm) centered at 289.2 nm ascended from the emission spectrum as the excitation power increased above the threshold pump power ($P_{th}$, around 0.28 J cm$^{-2}$). Moreover, the dependence of output intensity on the excitation power exhibited an "S" shape with three distinct regions (Fig. 5f), representing the transition from spontaneous emission through amplified spontaneous emission to gain saturation[48,49]. These results together confirm the onset of single-mode upconversion lasing.

We also demonstrated that the lasing features such as mode spacing, mode numbers, and threshold power can be precisely controlled by tuning the size of the microresonator (Supplementary Fig. S14). As the diameter of the microresonator increased, the number of lasing modes increased due to a decrease in the mode spacing (Fig. 5g, h). The observed mode spacing was well correlated with the parameters of the microresonators according to the following equation[49,50]:

$$\Delta\lambda = \lambda_0^2 / n_{eff} L \qquad (1)$$

where $\Delta\lambda$ is the mode spacing, $\lambda_0$ is the center peak wavelength, $n_{eff}$ (=1.52) is the effective refractive index and $L$ is the perimeter length of the microresonators. All of the above results confirm that, upon 1550 nm pumping, the toroidal microresonators were sufficient to create population inversion of a higher-lying excited state of Tm$^{3+}$ ions through DU for ultraviolet lasing. It is worth mentioning that multi-wavelength lasing action can be recorded at different emission peaks of the Tm$^{3+}$ and Er$^{3+}$ dopants (Supplementary Fig. S15). The remarkable tunability of lasing

emission in the judiciously designed NaYF$_4$:Yb/Tm@NaErF$_4$:-Ce@NaYF$_4$ nanoparticles certainly expands the possibility in future studies.

The 289 nm lasing from UCNPs-doped microresonator with an ultralarge anti-Stokes shift is susceptible to the Q-factor of the cavity, which enables sensitive detection of small biological species by monitoring the $P_{th}$ shift. As a proof of principle, we used a polystyrene (PS, 300 nm in diameter) sphere as the simulant of cancer cell secretion to conduct the sensing measurement. As anticipated, $P_{th}$ values of the 290 nm lasing increased considerably from 0.13 to 2.34 J cm$^{-2}$ by attaching a single PS sphere to the microresonator, due to the reduction of Q-factor from $2 \times 10^5$ to around $4 \times 10^4$ (Supplementary Fig. S16). The results demonstrate that our device integrating upconversion gain medium with high-Q microresonator structure is promising for designing high-quality sensing platforms.

## Discussion

In summary, we have established a DU scheme that allows upconverting excitation light in the telecommunication wavelength into deep-ultraviolet emissions with an ultralarge anti-Stokes shift of 1260 nm (~3.5 eV). The DU was realized by the cooperation of two distinct upconversion processes that are integrated into a single NaYF$_4$:Yb/Tm@NaErF$_4$:Ce@NaYF$_4$ nanoparticle through a Yb-mediated energy transfer at the core/shell interface. By using the DU nanoparticles as gain media, we further developed a novel toroid microcavity laser that manifested single-mode lasing at 289.2 nm. Our findings initiate an effective tactic to obtain upconversion lasers operating in the deep-ultraviolet regime by excitation at the telecommunication wavelength, which minimizes optical attenuation in SiO$_2$-based photonic circuits. Besides, the study of tandem combining different

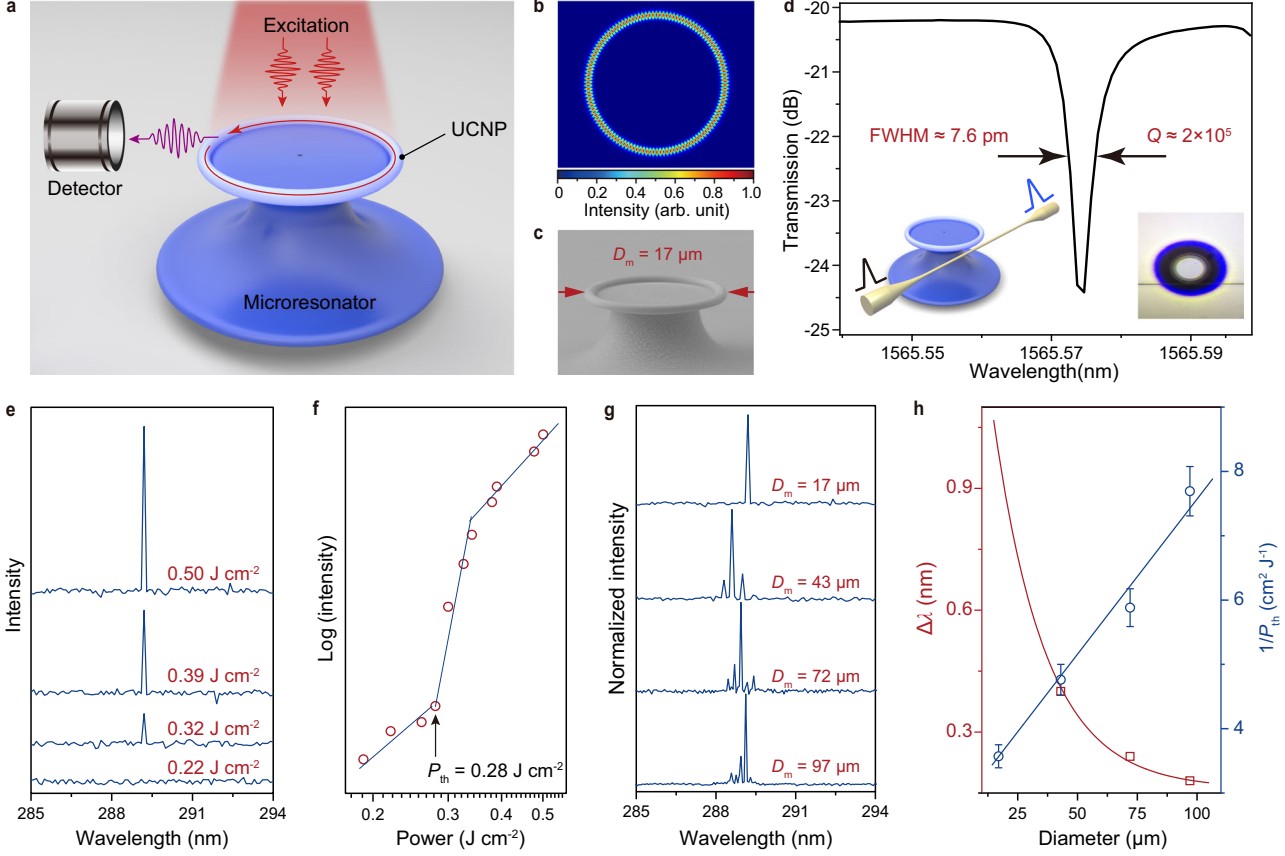

**Fig. 5 Deep-UV lasing in microresonator incorporated DU nanoparticles. a** Schematic setup of the microtoroidal resonator platform for upconversion lasing. **b** Simulation of the excited WGMs (given in 2D cross-sectional geometry) on the surface of the microcavity with a diameter of 4 μm and ring width of 0.2 μm, respectively. **c** SEM image of a typical UCNPs-doped microresonator. **d** The transmission spectrum of a UCNPs-doped microresonator, revealing a $Q$-factor of around $2 \times 10^5$. Inset: schematic of the measurement setup (left) and top-view photograph of the system under measurement (right). **e** Emission spectra of a microresonator with $D_m = 17$ μm at different excitation powers. **f** Logarithmic plot of output intensity versus excitation power for the microresonator. **g** Lasing spectra of microresonators with different $D_m$. **h** Plots of measured mode spacing ($\Delta\lambda$) and threshold pump power ($P_{th}$) of the microresonator as a function of $D_m$. Data points for threshold pump powers represent mean ± standard deviation (SD, $n = 3$). Error bars indicate SD.

upconversion processes through heavy lanthanide doping also raises new possibilities of constructing upconversion nanocrystals with highly tunable excitation and emission spectra for advanced biological and photonic applications.

## Methods

**Nanoparticle synthesis.** The multilayered $NaYF_4$:Yb/Tm@$NaErF_4$:Ce@$NaYF_4$ nanoparticles were synthesized according to the method in ref. [35]. Additional experimental details are provided in the Supplementary Information.

**Fabrication of toroid microcavity comprising upconversion nanoparticles.** A sol-gel silica film doped with upconversion nanoparticles was first made by using an acid-catalyzed hydrolysis-condensation reaction approach. Next, toroid microcavities were prepared from the as-synthesized upconversion sol-gel silica using a sequence of photolithography, etching, and laser-induced reflow. Additional experimental details are provided in the Supplementary Information.

**Theoretical modeling.** The electrical field in the waveguide structure was simulated by the three-dimensional finite-difference time-domain (3D-FDTD) method. The upconversion process in the $Er^{3+}$-$Ce^{3+}$ system was simulated by the rate equations of direct excitation and interionic cross-relaxation.

**Physical measurement.** HAADF-STEM images and HR-TEM images were measured with FEI Tecnai G2 F30 at 300 kV. The upconversion emission spectra were recorded with Ocean Optics USB 2000 and Maya 2000 PRO spectrometers. The lasing emission was measured by a monochromator (iHR-320) coupled with a photomultiplier tube. All measurements were performed at room temperature.

**Reporting summary**. Further information on research design is available in the Nature Research Reporting Summary linked to this article.

## Data availability

The data generated and analyzed during this study are available from the corresponding author upon reasonable request.

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

## Acknowledgements

This work was supported by the National Natural Science Foundation of China (Nos. 21773200, 21573185, 61805058, and 62105291), the Research Grants Council of Hong Kong (11205219, 11204717, and RFS2021-1S03), and Shenzhen Fundamental Research Fund (JCYJ20180306171700036).

## Author contributions

T.S., B.C., and F.W. conceived the projects, designed the experiments, and wrote the paper. T.S., B.C., Y.Guo, Q.Z., J.Z., X.C., and Y.Gao performed the experiments and analyzed the data. Y.L. and S.T.C. prepared the waveguide circuit. Y.W. and L.J. fabricated and tested the lasers. All authors contributed to the analysis of this manuscript.

## Competing interests

The authors declare no competing interests.
