## [Peer Review File · Nature Communications]

REVIEWER COMMENTS

Reviewer #1 (Remarks to the Author):

This work reports on the design and synthesis of a novel Ln doped nanostructure (with a complex and smart core/shell structure) that is capable of generating laser radiation in the UV under infrared (1.5 microns) optical excitation. Authors demonstrate the capability of their structure as nano laser generation is different configurations.

Since the recent demonstration of the potential of upconverting nanoparticles (UCNP) for laser generation, there is an increasing interest in this possibility. Once this possibility has been demonstrated, it is time to demonstrate the real application of such nanolasers. In the abstract and introduction authors claim the great potential of UV lasers for bio-applications. But the work later only deals with a nice description and characterization of the materials plus examples of laser oscillation. But nothing related to bio-applications. At present stage, the field needs not only good demonstrations of UCNP lasers but also requires evidences of their potential application. In this sense I think this is a nice paper with very good data and materials but its publication seems for me to be premature as the potential application of their nanolasers is not demonstrated.

At the end, it is not clear what are the benefits of shifting the laser wavelength down to the UV. For which applications is desirable to have 1.5 micron radiation as excitation source? I can not see the benefits of using this wavelength at the bio-world due to the string water absorption and tissue scattering at this wavelength.

This work has been submitted for publication in Nat Com, and I think authors should consider keep trying to publish it in Nat Com by adding some additional value to their work.

Reviewer #2 (Remarks to the Author):

In this manuscript the authors report about core-shell-shell UCNP, in which Er and Yb/Tm upconversion are separated in shell and core and the Er UC feeds the Yb/Tm UC with enhanced efficiency when Ce ions are added in the Er shell. The authors have performed many control experiments (ion per ion and at different concentrations, with different spacing, core-only NPs etc.) as well as different simulations, which make their experimental findings of sensitizing Tm via Er and Yb very convincing. By incorporating the UCNP into a whispering gallery mode microresonator, they even show the capability of UV lasing around 290 nm, which is a very impressive result. The results are important and appealing to a broad audience. Also, they offer quite some space for further investigation (in particular for the lasing part, which is quite short). I also found the manuscript well written and most of the different experiments and results well explained. I recommend the publication of this manuscript in Nature Communications and have only a few comments that the authors can take into account for improving the clarity of some parts of their study.

Comments:

P1: What are "ionic levels"?

P1: What are "enormous applications"?

Why do the authors call the process "domino upconversion"? In my understanding, upconversion is always a multiphoton process, which can comprise 2 or more photons and the sequential absorption of photons by different ions.

Figure 1 can be erased because it is already shown in more detail in Figure 2a. Moreover, in Figure 1 the principle goes from left to right, whereas in Figure 2a it goes from right to left (that may cause some additional confusion).

In SuppFig4b it would be very helpful (for better direct comparison) to also include the spectrum of

NaYF₄:Yb/Tm@NaErF₄:Ce@NaYF₄.

Figure 3a: Are these spectra somehow intensity-normalized? Otherwise, how can the authors be sure that the different NPs can be directly compared?

SuppFigure 6b: It is a bit difficult to distinguish the different spectra (in particular for the Tm emission. Also, and even more important, the 30/70 case does not show the same results as in Figure 3a (there is substantial 290 emission in Fig3a but nothing - only for the 80/20 case - in SuppFig6b). This difference should be explained.

Fig4b: The colors (all blue) are really difficult to distinguish. I think no spacing is the best but I am not 100% sure.

How was the lasing exactly measured? Was the entire ring measured from the top (as shown in the inset in Fig5d) or is the lasing light coupled out as some specific part of the ring (as shown in Fig5a)? That may be a stupid question for someone who is very familiar with such ring resonators but the broader audience may not understand it.

Why is there lasing only at the 290nm emission line of Tm. All other emission lines are much stronger and there should be lasing from those lines also (and even at lower pump energy). That part of the manuscript (lasing) should be treated with more detailed explanations.

Point-by-Point Response to Referees

We want to thank the referees for their careful reading of our manuscript and the thoughtful comments provided, which helped us further enrich our manuscript. In response to the points raised by the referees (marked in black), we provide point-by-point responses (marked in blue) along with the modifications (marked in red) made in the revised manuscript.

Referee #1

1. This work reports on the design and synthesis of a novel Ln doped nanostructure (with a complex and smart core/shell structure) that is capable of generating laser radiation in the UV under infrared (1.5 microns) optical excitation. Authors demonstrate the capability of their structure as nano laser generation is different configurations.

Since the recent demonstration of the potential of upconverting nanoparticles (UCNP) for laser generation, there is an increasing interest in this possibility. Once this possibility has been demonstrated, it is time to demonstrate the real application of such nanolasers. In the abstract and introduction authors claim the great potential of UV lasers for bio-applications. But the work later only deals with a nice description and characterization of the materials plus examples of laser oscillation. But nothing related to bio-applications. At present stage, the field needs not only good demonstrations of UCNP lasers but also requires evidences of their potential application. In this sense I think this is a nice paper with very good data and materials but its publication seems for me to be premature as the potential application of their nanolasers is not demonstrated.

Response: We thank the referee for the valuable comments, and we also appreciate the recognition from the referee for the realization of UCNP-based UVB laser in our device. High Q-factor microcavities, confining photons in a small volume for a long duration time, can significantly enhance light-matter interactions, therefore making it a suitable platform for revolutionizing bio-sensing and point-of-care diagnosis [Science 2007, 317, 783; Nature Nanotechnology 2014, 9, 933]. With the increasing demands in the early detection of cancers, the past decades have witnessed the development of optical biosensors based on high Q-factor whispering gallery mode (WGM) microcavities. Such WGM-based sensors have been reported for the detection of biomolecules and nanoparticles by monitoring different parameters such as wavelength shift, mode splitting, and mode broadening [PNAS 2008, 105, 20701; Nature Photonics 2010, 4, 46; Nature Nanotechnology 2011, 6, 428]. However, those sensing strategies, obtained from the transmission or reflection of the target resonators, highly rely on ultrahigh Q ($>10^8$) resonators [Nature Photonics 2010, 4, 46; Physical Review A 2007, 76, 013823; Nature Methods 2008, 5, 591], which imposes challenges in fabrication. In addition, undesired signal noise, arising from laser instability, mechanical vibration and temperature

fluctuation, remains another obstacle for further application [Lab on a Chip 2017, 17, 1190; Optics Express 2014, 22, 3098].

To show the potential of our upconversion lasers, we exploit a robust sensing mechanism to detect small biological species by monitoring the emission spectra. In contrast to conventional coupling systems using tapered fiber or waveguides, this strategy is realized through the free-space excitation and detection, which facilitates the integration with microfluidics chips for potential *in vivo* characterizations. Intuitively, the destructive interference caused by the adsorption of the small species leads to a significant Q-degradation of the resonator, therefore increasing the threshold value (P_{th}). As a proof of concept, we used a polystyrene (PS, 300 nm in diameter) sphere to simulate cancer cell secretions as a model analyte.

The optical images in Supplementary **Figure 15a** display our experimental setup. By plotting the simulated field patterns of the excited microresonator with/without PS sphere (Supplementary **Figure 15b,c**), we found that deleterious interaction occurs at the joint position between the resonant modes and the target sphere, accompanied by a light escape out of the volume. The broken total internal reflection has been experimentally verified in the microlasers, as shown in Supplementary **Figure 15d,e**. As anticipated, P_{th} values of 290 nm lasing increased considerably from 0.13 to 2.34 J cm⁻² by attaching a PS sphere to the microresonator (Supplementary **Figure 15f**). This is consistent with the significant Q-factor reduction (from 2×10^5 to around 4×10^4) deduced from the linewidth broadening, as shown in Supplementary **Figure 15g**.

Accordingly, we added the following sentences to the revised manuscript. We hope the referee agrees.

*The 289 nm lasing from UCNPs-doped microresonator with an ultralarge anti-Stokes shift is susceptible to the Q-factor of the cavity, which enables sensitive detection of small biological species by monitoring the P_{th} shift. As a proof of principle, we used a polystyrene (PS, 300 nm in diameter) sphere as the simulant of cancer cell secretion to conduct the sensing measurement. As anticipated, P_{th} values of the 290 nm lasing increased considerably from 0.13 to 2.34 J cm⁻² by attaching a single PS sphere to the microresonator, due to the reduction of Q-factor from 2×10^5 to around 4×10^4 (Supplementary **Fig. S15**). The results demonstrate that our device integrating upconversion gain medium with high-Q microresonator structure is promising for designing high-quality sensing platforms.*

Supplementary Figure 15. A sensing platform based on the UCNP-doped toroidal microlaser by monitoring the P_{th} change. (a) Optical image of a PS sphere (300 nm in diameter) attached to the microresonator structure ($\sim 100 \mu\text{m}$ in diameter). The white arrow indicates the PS sphere. (b, c) The simulated field patterns of the excited microresonator with and without PS sphere, respectively. In the calculation, the diameter/width of the microring and the diameter of the PS sphere were set at $4/0.2 \mu\text{m}$ and $0.2 \mu\text{m}$, respectively. The results reveal that deleterious interaction occurs at the joint position between the resonant modes and the target sphere, accompanied by a light escape out of the volume. (d, e) Logarithmic plot of output intensity versus excitation power for the microresonator without and with PS sphere, respectively. (f) Statistics of P_{th} values of six microresonators without (bottom panel) and with a PS sphere attached at different points (top panel). (g) Transmission spectra from the UCNP-doped microresonator attached with an external 300-nm-diameter PS sphere.

2. At the end, it is not clear what are the benefits of shifting the laser wavelength down to the UV. For which applications is desirable to have 1.5 micron radiation as excitation source? I can not see the benefits of using this wavelength at the bio-world due to the strong water absorption and tissue scattering at this wavelength.

Response: Our platform displays an upconversion lasing with an ultralarge anti-Stokes shift. Using upconversion lasing as an excitation source, we design a detecting system that encompasses the benefits of low background noise (defined by the upconversion mechanism) and high sensitivity (arising from the multiphoton upconversion process and high-Q value of the resonator), which are critical for sensing application.

The benefit of lasing with wavelength down to UV by excitation at telecommunication wavelengths is promising, partially benefiting from its large anti-Stokes shift. Other applications can be expected such as microbial sterilization and biomedical instrumentation systems as we mentioned in the introduction section. The 1.5 micron radiation at NIR III (1550-1800 nm) is suitable for fiber-optic communication and photonic circuits because of minimal optical attenuation in the SiO₂ micro-resonator [Willner, Alan. Optical fiber telecommunications. Vol. 11. Academic Press, 2019.]. Besides, the water absorption and tissue scattering of the excitation source can be neglected in our design because the excitation pathway is free of water for the lasing generation.

We added the following sentence to the revised manuscript.

Our findings initiate a novel tactic to obtain upconversion lasers operating in the deep ultraviolet regime by excitation at the telecommunication wavelength, which minimizes optical attenuation in SiO₂-based photonic circuits.

We hope the referee agrees.

3. This work has been submitted for publication in Nat Com, and I think authors should consider keep trying to publish it in Nat Com by adding some additional value to their work.

Response: We would like to express our gratitude to the referee for his/her positive remarks again. We believe that our revised manuscript is now suitable for publication in Nat Commun.

Referee #2

In this manuscript the authors report about core-shell-shell UCNPs, in which Er and Yb/Tm upconversion are separated in shell and core and the Er UC feeds the Yb/Tm UC with enhanced efficiency when Ce ions are added in the Er shell. The authors have performed many control experiments (ion per ion and at different concentrations, with different spacing, core-only NPs etc.) as well as different simulations, which make their experimental findings of sensitizing Tm via Er and Yb very convincing. By incorporating the UCNPs into a whispering gallery mode microresonator, they even show the capability of UV lasing around 290 nm, which is a very impressive result. The results are important and appealing to a broad audience. Also, they offer quite some space for further investigation (in particular for the lasing part, which is quite short). I also found the manuscript well written and most of the different experiments and results well explained. I recommend the publication of this manuscript in Nature Communications and have only a few comments that the authors can take into account for improving the clarity of some parts of their study.

Response: We thank the referee for his/her careful reading of our manuscript and the constructive remarks.

1. What are “ionic levels”?

Response: We apologies for any confusion. We should have referred to “the energy levels of lanthanide ions”. Therefore, we have changed the “ionic levels” to “energy levels” in the revised manuscript.

2. What are “enormous applications”?

Response: The deep UV emission has played an essential role in medical and industrial applications such as microbial sterilization and biomedical instrumentation systems. For example, Xia and co-workers utilized the deep UV emission of $\text{Li}_2\text{GaGeO}_4:\text{Pr}^{3+}$ for microbial sterilization application [Sci. China Mater. 2021, doi:10.1007/s40843-021-1790-1]; Kim and co-workers proposed a concept of upconversion antimicrobial surface based on the deep UV emission of $\text{Y}_2\text{SiO}_5:\text{Pr}^{3+}$ [Environ. Sci. Technol. 2012, 46, 12316]. In addition, deep UV emission is also useful for constructing advanced fluorescence imaging microscopy systems. For example, Kawata and co-workers used deep UV resonant Raman spectroscopy for photodamage characterization in cells [Biomed. Opt. Express 2011, 2, 927]; Juodkazis and co-workers developed a deep UV fluorescence lifetime imaging microscopy for recognition of proteins [Photon. Res. 2015, 3, 283]. The proper references have been added to the revised manuscript.

To be more informative, we also modified the following sentences in the revised manuscript.

“Frequency upconversion holds potential for cost-effective construction of miniaturized deep-UV emission devices that find enormous medical and industrial applications, such as microbial sterilization and biomedical instrumentation.”

3. Why do the authors call the process “domino upconversion”? In my understanding, upconversion is always a multiphoton process, which can comprise 2 or more photons and the sequential absorption of photons by different ions.

Response: To make our idea more concise, we revised **Figure 1** by comparing a conventional energy transfer upconversion (ETU) process and the proposed domino upconversion (DU) process, as shown below. We coined the special upconversion process in our manuscript as “domino upconversion” because it consists of a tandem combination of distinct upconversion processes that are integrated into separate layers of the core-shell-shell structure. In a conventional upconversion process, the excitation energy is only amassed in one type of upconverting lanthanide ions such as Er^{3+} (**Figure 1a**). While in our proposed “domino upconversion” process, domino means that the

energy amassed in one upconversion course (i.e., Er^{3+} -based upconversion) triggers a succeeding upconversion process (i.e., Tm^{3+} -based upconversion) (**Figure 1b**).

We hope now it is now clear for reading.

Revised Figure 1 | Comparison of the conventional energy transfer upconversion (ETU) and the proposed domino upconversion (DU) processes. **a**, In an ETU process, excitation energy is only amassed in one type of lanthanide upconverting ion. **b**, In a DU process, excitation energy amassed in one upconverting ion triggers energy amassment in a second upconverting ion, leading to an ultralarge anti-Stokes shift.

4. Figure 1 can be erased because it is already shown in more detail in Figure 2a. Moreover, in Figure 1 the principle goes from left to right, whereas in Figure 2a it goes from right to left (that may cause some additional confusion).

Response: We have thought over the referee's suggestion. Actually, we intend to highlight the idea of DU process, which is the central concept of our manuscript. As **Figure 2a** is the specific case of DU, we want to persuade the referee to keep **Figure 1** in our manuscript. In addition, to address the confusion mentioned by the referee, we revised **Figure 1** by exchanging the order from right to left now. We anticipate that the modification we made can address the referee's confusion, and we hope the referee agrees.

5. In SuppFig4b it would be very helpful (for better direct comparison) to also include the spectrum of $\text{NaYF}_4:\text{Yb}/\text{Tm}@\text{NaErF}_4:\text{Ce}@\text{NaYF}_4$.

Response: According to the referee's suggestion, we have included the spectrum of $\text{NaYF}_4:\text{Yb}/\text{Tm}@\text{NaErF}_4:\text{Ce}@\text{NaYF}_4$ in the revised Supplementary **Figure 4b**. We also provide the corresponding TEM images of these samples for references in the revised Supplementary **Figure 4a**.

Revised Supplementary Figure 4. Characterization of the $\text{NaYF}_4:\text{Yb/Tm}@NaYF_4$, $\text{NaYF}_4@NaYF_4:\text{Er}@NaYF_4$, and $\text{NaYF}_4:\text{Yb/Tm}@NaErF_4:\text{Ce}@NaYF_4$ nanoparticles. (a) TEM images of $\text{NaYF}_4:\text{Yb/Tm}@NaYF_4$, $\text{NaYF}_4@NaYF_4:\text{Er}@NaYF_4$, and $\text{NaYF}_4:\text{Yb/Tm}@NaErF_4:\text{Ce}@NaYF_4$ nanoparticles, respectively. Scale bars are 50 nm. (b) Emission spectra of $\text{NaYF}_4:\text{Yb/Tm}@NaYF_4$, $\text{NaYF}_4@NaYF_4:\text{Er}@NaYF_4$, and $\text{NaYF}_4:\text{Yb/Tm}@NaErF_4:\text{Ce}@NaYF_4$ nanoparticles under 1550 nm excitation, respectively. Note that the emission spectrum of $\text{NaYF}_4@NaYF_4:\text{Er}@NaYF_4$ nanoparticles was used for reference. The absence of emission peaks from the $\text{NaYF}_4:\text{Yb/Tm}@NaYF_4$ core-shell nanoparticle indicated that the Yb/Tm-doped upconversion layer does not respond to the 1550 nm excitation.

6. Figure 3a: Are these spectra somehow intensity-normalized? Otherwise, how can the authors be sure that the different NPs can be directly compared?

Response: The spectra recorded in **Figure 3a** are not intensity-normalized spectra. Those are absolute spectra and the intensity of each sample can be directly compared. These three batches of $\text{NaYF}_4:\text{Yb/Tm}@NaYF_4:\text{Ce/Er}$ (x/y%) $@NaYF_4$ nanoparticle were made by strictly following the same synthetic protocols. And most importantly, in order to minimize undesired variations caused by different batches of core and outermost shell precursor, the same batch of $\text{NaYF}_4:\text{Yb/Tm}$ core nanoparticle and NaYF_4 shell precursor were respectively utilized during the synthesis of each set of nanoparticles. We also made sure that the excitation and emission measurement setup remained the same. Therefore, we reason that the emission spectra in **Figure 3a** can be directly compared.

7. SuppFigure 6b: It is a bit difficult to distinguish the different spectra (in particular for the Tm emission). Also, and even more important, the 30/70 case does not show the same results as in Figure 3a (there is substantial 290 emission in Fig3a but nothing - only for the 80/20 case - in SuppFig6b). This difference should be explained.

Response: We thank the referee for raising this critical issue. In the revised manuscript, we modified Supplementary **Figure 6** by changing the line colors and emphasizing the emission in the range of 250-370 nm as an inset graph for direct comparison.

Regarding the referee's second concern, the data of 30/70% Ce/Er nanoparticle in Figure 3a and Figure S6b are, in fact, the same. Although the nanoparticle with 30/70% Ce/Er doping concentration showed the highest 290 nm emission, the emissions at longer wavelengths for this sample are much lower than nanoparticles with less Ce doping concentration such as Ce/Er (0/100%). I think this may be the reason that misled the referee.

After we change the color with high contrast, we hope it is now clear for reading.

Revised supplementary Figure 6

8. Fig4b: The colors (all blue) are really difficult to distinguish. I think no spacing is the best but I am not 100% sure.

Response: We thank the referee for the careful reading of our manuscript. We have changed the line colors with a higher contrast.

Revised Figure 4

9. How was the lasing exactly measured? Was the entire ring measured from the top (as shown in the inset in Fig5d) or is the lasing light coupled out as some specific part of the ring (as shown in Fig5a)? That may be a stupid question for someone who is very familiar with such ring resonators but the broader audience may not understand it.

Response: We appreciate the referee for the valuable comment. For lasing characterization, a 1550 nm pulsed laser (pulse width 6 ns, repetition rate 10 Hz, Φ 8 mm) is directly focused onto the top surface of the UCNPs-doped microresonator. The emission light from the boundary of the cavity is collected by an optical fiber coupled to an iHR-320 (Horiba) monochromator attached with a photomultiplier tube. This experimental setup is schematically shown in the inset of Fig. 5a. In addition, Q-factor is given to demonstrate the interface quality of the microresonator since whispering gallery modes occur near the interface of the volume with its surroundings. As shown in the inset of Fig. 5d, the measurement was conducted in the telecommunication band (~1550 nm). In the coupling system, a single-mode semiconductor tunable laser was used to excite the microresonator through a tapered fiber. Note that the waist of the tapered fiber is ~1 μ m. The axial direction of the fiber was kept along with the equatorial plane of the resonator. By scanning the pumping wavelengths, the transmission spectra were recorded by an

oscilloscope. A typical transmission spectrum for a 97- μm -diameter microresonator structure is depicted in **Fig. 5d**. Following the equation $Q = \lambda/\delta\lambda$ (λ and $\delta\lambda$ are the resonant wavelength and the corresponding full width of half maximum, respectively), we observe Q values of over 10^5 for the UCNPs-doped microresonator.

To be informative, we added the statement in Supplementary Information in the revised manuscript as below. We hope our description can help the referee and other readers to understand our setup.

11. Lasing characterization. *A 1550 nm pulsed laser (pulse width 6 ns, repetition rate 10 Hz, $\Phi 8$ mm) was directly focused onto the top surface of the UCNPs-doped microresonator. The emission light from the boundary of the cavity was collected by an optical fiber coupled to an iHR-320 (Horiba) monochromator attached with a photomultiplier tube. The Q -factor was given to demonstrate the interface quality of the microresonator since whispering gallery modes occur near the interface of the volume with its surroundings. In the coupling system, a single-mode semiconductor tunable laser was used to excite the microresonator through a tapered fiber. Note that the waist of the tapered fiber is ~ 1 μm . The axial direction of the fiber was kept along with the equatorial plane of the resonator. By scanning the pumping wavelengths, the transmission spectra were recorded by an oscilloscope. Following the equation $Q = \lambda/\Delta\lambda$ (λ and $\Delta\lambda$ are the resonant wavelength and the corresponding full width of half maximum, respectively). We observe Q values of over 10^5 for a typical UCNPs-doped microresonator.*

10. Why is there lasing only at the 290nm emission line of Tm. All other emission lines are much stronger and there should be lasing from those lines also (and even at lower pump energy). That part of the manuscript (lasing) should be treated with more detailed explanations.

Response: We appreciate this valuable comment raised by the referee. We agree with the referee that multi-wavelength lasing can be obtained through the UCNPs-based microresonator. In response to the referee's questions, we plotted the multi-wavelength lasing spectra from the transitions of Er^{3+} and Tm^{3+} ions in our revised **Supplementary Figure 14**.

In **Supplementary Figure 14**, we summarize the multi-wavelength lasing action in a UCNPs-based microresonator (~ 100 μm in diameter) under the excitation of a 1550 nm pulsed laser. All the emission peaks can be attributed to the Er^{3+} transitions (i.e., 381 nm, 410 nm, 526 nm, and 547 nm, respectively) and Tm^{3+} transitions (i.e., 289 nm, 346 nm, 362 nm, 450 nm, and 475 nm, respectively). For instance, at the very beginning, only a broad emission band centered at 381 nm emerged from the spectra. With the increase of pumping power, sharp peaks with periodic mode spacing ($\Delta\lambda$) ascended and quickly dominate the emission spectra above P_{th} , as reflected by the kink value of the light-light curve in **Supplementary Figure 14b,c**. In addition, the experimental $\Delta\lambda$ was found to be

around 0.30 nm, which is consistent with the calculated one following the equation $\Delta\lambda = \lambda_0^2/n_{\text{eff}}L$ ($n_{\text{eff}} \approx 1.52$). Hence, all these results clearly evidence the whispering gallery mode (WGM)-based lasing emission. Similarly, multimode WGM-based lasing behavior can be observed in other characteristic emission bands.

We added the following sentences in our revised manuscript.

It is worth mentioning that multi-wavelength lasing action can be recorded at different emission peaks of the Tm^{3+} and Er^{3+} dopants (Supplementary Fig. S14). The remarkable tunability of lasing emission in the judiciously designed $\text{NaYF}_4:\text{Yb}/\text{Tm}@/\text{NaErF}_4:\text{Ce}@/\text{NaYF}_4$ nanoparticles certainly expands the possibility in future studies.

Supplementary Figure 14. Multi-wavelength lasing from the transitions of Er^{3+} and Tm^{3+} ions in a UCNPs-doped microresonator ($\sim 100 \mu\text{m}$ in diameter). (a) The simplified energy levels showing the transitions from Er^{3+} and Tm^{3+} ions. (b, c) Logarithmic plot of the output intensities of emissions from Er^{3+} and Tm^{3+} ions versus excitation power for the microresonator, respectively. (d-f) The power-dependent emission spectra at the wavelengths of 381 nm, 410 nm, 526 nm and 547 nm for Er^{3+} ions, respectively. (g-i) The power-dependent emission spectra at the wavelengths of 289 nm, 346 nm, 362 nm, 450 nm, and 475 nm for Tm^{3+} ions, respectively. Note that the lasing thresholds are identified to be P_{th} (289 nm) = 0.13 J cm^{-2} , P_{th} (346 nm) = 0.13 J cm^{-2} , P_{th} (362 nm) = 0.11 J cm^{-2} , P_{th} (450 nm) = 0.11 J cm^{-2} , P_{th} (475 nm) = 0.08 J cm^{-2} , P_{th} (381 nm) = 0.11 J cm^{-2} , P_{th} (410 nm) = 0.10 J cm^{-2} , P_{th} (526 nm) = 0.08 J cm^{-2} , and P_{th} (547 nm) = 0.07 J cm^{-2} , respectively.

REVIEWERS' COMMENTS

Reviewer #1 (Remarks to the Author):

I have revised the amended version of the manuscript entitled "Ultralarge Anti-Stokes Lasing through Domino Upconversion". In my previous report I was asking authors to demonstrate a practical application of their photonic structures for sensing. It is true that they have included some proof of concept experiments that demonstrate the presence of an attached microsphere. The detection parameter is the laser threshold. Optical detection based on the measurement of the laser threshold is not a realistic procedure for detection as it implies measuring laser curves (time consuming and requiring complicated optical system). So, in my opinion, this proof of concept does not really demonstrate the potential of these photonic structures for sensing. So, in this sense, I think the paper has not really been improved. Being said that, I think this is an editorial decision. The paper is a nice one, fundamental results are nice but in my opinion the paper lacks a demonstration of the potential applications. If you decide to publish it I would understand. My reasoning is not enough to reject the paper. I would be happy if authors could include some bio-sensing experiments (or even in microfluidics).

Reviewer #2 (Remarks to the Author):

The authors have carefully revised the manuscript, replied to all comments of both reviewers and have also added new data (mainly in the supporting information) and explanations. I still find the study very interesting and I recommend to publish the revised manuscript in Nature Communications. If there is enough space, I also agree in keeping Figure 1 (I leave this decision to the editor). I agree with Reviewer 1 that an actual biological application would be great. However, I think the presented results alone are important enough for publication in Nature Communications and hopefully the results will stimulate further research to see bioapplications in the near future.

Point-by-Point Response to Referees

We thank the referees for their contribution to improving our manuscript. In response to the points raised by the referees, we here provide point-by-point responses.

Referee #1

I have revised the amended version of the manuscript entitled "Ultralarge Anti-Stokes Lasing through Domino Upconversion". In my previous report I was asking authors to demonstrate a practical application of their photonic structures for sensing. It is true that they have included some proof of concept experiments that demonstrate the presence of an attached microsphere. The detection parameter is the laser threshold. Optical detection based on the measurement of the laser threshold is not a realistic procedure for detection as it implies measuring laser curves (time consuming and requiring complicated optical system). So, in my opinion, this proof of concept does not really demonstrate the potential of these photonic structures for sensing. So, in this sense, I think the paper has not really been improved.

Being said that, I think this is an editorial decision. The paper is a nice one, fundamental results are nice but in my opinion the paper lacks a demonstration of the potential applications. If you decide to publish it I would understand. My reasoning is not enough to reject the paper. I would be happy if authors could include some bio-sensing experiments (or even in microfluidics).

Response: We thank the referee for the deep insight comment. The main comment is the potential bio-application of the photonic device. As you know, with the common goal towards real-world sensing applications, high Q-factor microcavities have been employed with extraordinary success in the label-free detection of small species. Such advances come from the fact that the transmission/reflection spectrum of silica-based microcavity can produce significant mode splitting, resonance shift, and mode broadening phenomena with high sensitivity. Soon after, with the improvement in cavity design, the sensors can identify specific adsorption species in complex media.

In this manuscript, the proposed upconversion nanoparticles with ultralarge anti-Stokes shift were incorporated in the chip-integrated microtoroid cavity for sensing application by monitoring the lasing threshold. It is worth noting that instead of measuring the lasing threshold changes, the nanoparticle adsorption event can be revealed in real time by monitoring the sharp fall in emission intensity when the excitation power locates between P_1' and P_2 . (P_1' and P_2 represent the gain saturation point of the microtoroid attached without /with a 300-nm PS sphere). Therefore, we believe that this simplified strategy could facilitate qualitative measurements outside the laboratory. In this regard, our paper has indeed been improved. We hope that the referee concurs.

Referee #2

The authors have carefully revised the manuscript, replied to all comments of both reviewers and have also added new data (mainly in the supporting information) and explanations. I still find the study very interesting and I recommend to publish the revised manuscript in Nature Communications. If there is enough space, I also agree in keeping Figure 1 (I leave this decision to the editor).

I agree with Reviewer 1 that an actual biological application would be great. However, I think the presented results alone are important enough for publication in Nature Communications and hopefully the results will stimulate further research to see bioapplications in the near future.

Response: We appreciate the referee for recognizing the novelty of our research and the suggestion of acceptance. We would like to express our gratitude for your careful review and valuable suggestions that have helped us improve the quality of our manuscript.